# USP38 Inhibits Zika Virus Infection by Removing Envelope Protein Ubiquitination

**DOI:** 10.3390/v13102029

**Published:** 2021-10-08

**Authors:** Yingchong Wang, Qin Li, Dingwen Hu, Daolong Gao, Wenbiao Wang, Kailang Wu, Jianguo Wu

**Affiliations:** 1State Key Laboratory of Virology, College of Life Sciences, Wuhan University, Wuhan 430072, China; 2016202040049@whu.edu.cn (Y.W.); 2019202040052@whu.edu.cn (Q.L.); 2018202040031@whu.edu.cn (D.H.); wukailang@whu.edu.cn (K.W.); 2Guangdong Longfan Biological Science and Technology Company, Shunde District, Foshan 528315, China; gdl@gd-longfan.com; 3Guangdong Provincial Key Laboratory of Virology, Institute of Medical Microbiology, Jinan University, Guangzhou 510632, China; shabiao1212@whu.edu.cn; 4Foshan Institute of Medical Microbiology, Foshan 528315, China

**Keywords:** deubiquitinase, envelope protein, USP38, virus infection, Zika virus

## Abstract

Zika virus (ZIKV) is a mosquito-borne flavivirus, and its infection may cause severe neurodegenerative diseases. The outbreak of ZIKV in 2015 in South America has caused severe human congenital and neurologic disorders. Thus, it is vitally important to determine the inner mechanism of ZIKV infection. Here, our data suggested that the ubiquitin-specific peptidase 38 (USP38) played an important role in host resistance to ZIKV infection, during which ZIKV infection did not affect USP38 expression. Mechanistically, USP38 bound to the ZIKV envelope (E) protein through its C-terminal domain and attenuated its K48-linked and K63-linked polyubiquitination, thereby repressed the infection of ZIKV. In addition, we found that the deubiquitinase activity of USP38 was essential to inhibit ZIKV infection, and the mutant that lacked the deubiquitinase activity of USP38 lost the ability to inhibit infection. In conclusion, we found a novel host protein USP38 against ZIKV infection, and this may represent a potential therapeutic target for the treatment and prevention of ZIKV infection.

## 1. Introduction

Zika virus (ZIKV) was first isolated from rhesus monkeys in the Zika forest of Uganda, Africa, in 1947 [1]. The viral infection was originally thought to be mild and have limited effect on humans [2]. An epidemic of ZIKV in Asia and the Pacific Islands was reported in 2007 [3], followed by an outbreak in French Polynesia in 2013–2014 [4]. By 2015, there had been a large-scale outbreak of ZIKV infection in South America [5]. By 2017, more than 220,000 confirmed cases had been reported in 52 countries or regions in the Americas, which aroused global attention [6]. ZIKV is an enveloped single-stranded, positive-sense RNA virus belongs to the family Flaviviridae [7], which contains the hepatitis C virus (HCV), Japanese encephalitis virus (JEV), West Nile virus (WNV), and dengue virus (DENV). ZIKV is a mosquito-borne virus, and methods of its transmission also include blood transfusion, mother-to-child, and sexual transmission [8,9,10]. ZIKV has a positive single-stranded RNA genome about 11 kb, encoding seven nonstructural (NS) proteins: NS1, NS2A, NS2B, NS3, NS4A, NS4B, and NS5, as well as three structural proteins: capsid (C), premembrane/membrane (prM), and envelope (E) [11,12]. ZIKV infection has a major impact on human health, and several research works revealed that ZIKV infection is involved in several neurological disorders, such as Guillain-Barré and/or microcephaly [13,14].

Ubiquitin-specific peptidase 38 (USP38) is a member of ubiquitin-specific processing enzymes family, and it has been reported that it inhibits type I IFN signaling during viral infection [15], which is also associated with some diseases, such as ciliopathies and asthma [16,17]. It has been shown that USP38 affects DNA damage repair by regulating the activity of HDAC1, meanwhile, a low expression of USP38 causes genome instability, which may lead to tumorigenesis [18]. USP38 is also a regulator of histone modification and inflammation [19].

The envelope of mature ZIKV is composed of the E protein and M protein, and the main function of ZIKV E protein is receptor binding, membrane fusion, and host cell entry [20,21,22,23], in addition to antibody neutralization [24]. Post-translational modifications of E protein are essential for its functioning. This includes ubiquitination and glycosylation of the E protein, which is necessary for ZIKV invasion and pathogenesis [25,26,27]. Specifically, it was previously shown that K63-linked polyubiquitination of E is important for virus attachment and can be a determinant of tissue tropism [25].

In this study, our results revealed that overexpression of USP38 in Hela cells inhibited ZIKV invasion, and that Hela cells stably expressing USP38 shRNA significantly promoted ZIKV invasion. ZIKV infection had no effect on USP38 expression in Hela cells. Furthermore, USP38 could interact with the ZIKV E protein and co-localize with ZIKV E protein in the cytoplasm. Meanwhile, USP38 decreased E protein polyubiquitination. At the same time, the deubiquitinase activity of USP38 was indispensable for inhibiting the replication of ZIKV. In conclusion, we revealed a new function of USP38 in viral infection and this may provide a new potential therapeutic target for ZIKV-associated diseases.

## 2. Materials and Methods 

### 2.1. Cell Lines and Cultures

HEK293T, Hela (ATCC, #CCL-2), A549 (ATCC, #CCL-185) and C6/36 (ATCC, #CRL-1660) cells were purchased from the American Type Culture Collection (ATCC, Manassas, VA, USA). HEK293T cells, A549 cells, and Hela cells were cultured in Dulbecco’s modified Eagle’s medium (DMEM) (Gibco, Grand Island, NY, USA) supplemented with 10% FBS, 100 U/mL penicillin, and 100 μg/mL streptomycin sulfate. C6/36 cells were cultured in RPMI 1640 medium (Gibco) supplemented with 10% FBS, 100 U/mL penicillin, and 100 μg/mL streptomycin sulfate. HEK293T and Hela cells were maintained in an incubator at 37 °C with 5% CO_2_. C6/36 cells were maintained in an incubator at 30 °C with 5% CO_2_. 

### 2.2. Antibodies and Reagents

Antibody against Flag (F3165) (1:1000), HA (H6908) (1:2000), and GAPDH (G9295) (1:5000) were purchased from Sigma (St Louis, MO, USA). Antibody against Myc (2276S) (1:2000) was purchased from Cell Signaling Technology (CST, Boston, MA, USA). Antibodies against ZIKV NS5 protein (GTX133312) (1:1000) and Envelope protein (GTX133314) (1:1000) were purchased from GeneTex (Hsinchu City, Taiwan, P.R.C). Antibody targeting USP38(ab72244) (1:1000) was purchased from Abcam (Shanghai, Hubei, P.R.C). PEI was purchased from Sigma (St Louis, MO, USA). RPMI-1640 and DMEM were purchased from Gibco (Grand Island, NY, USA). Protein ladder (26616) was purchased from Thermo Scientific (Rockford, IL, USA). Complete, EDTA-free Protease Inhibitor Cocktail was purchased from Roche (Basel, Switzerland). Normal rabbit immunoglobulin G (IgG), and normal mouse immunoglobulin G were purchased from Invitrogen Corporation (Carlsbad, CA, USA).

### 2.3. Western Blotting and Co-Immunoprecipitation Assay

Cells were lysed at 4 °C with lysis buffer in the presence of cocktail (Roche), centrifuged at 4 °C at 12,000 rpm for 10 min to collect the supernatant, mixed with the loading buffer, and then subjected to 8% or 10% SDS-PAGE and Western blotting. For Co-IP analysis, 50 µL of supernatant after centrifugation was used for Western blotting. The left lysate was incubated with the indicated primary antibody overnight. The next day, the mixture was incubated with A/G-Sepharose for 2 h and then added to the loading buffer for Western blot analysis. The formula of the lysis buffer used in this study is as follows: 50 mM Tris-HCl, pH7.4–7.6, 300 mM NaCl, 1% Triton-X, 5 mM EDTA, and 10% glycerol. 

### 2.4. Immunofluorescence Analysis

After being transfected with plasmids or infected with the virus as indicated, Hela cells were washed three times with PBS, then fixed in 4% paraformaldehyde, permeabilized with 0.1% Triton X-100 and blocked with 5% BSA, and finally the cells were incubated with the primary antibody overnight at 4 °C. The next day, the secondary antibodies and DAPI were added to the cells after washing off the primary antibodies. The cells were analyzed using a confocal laser scanning microscope (Fluo View FV1000; Olympus, Tokyo, Japan).

### 2.5. Quantitative RT-PCR Analysis and Primers

RNA is extracted by Ultrapure RNA Kit (CWBIO), cDNA is reverse transcribed by HiScript II Reverse Transcriptase (Vazyme). All specific primers for testing were designed by Primer Bank or Primer Premier 5.0. The primer sequences were as follows: ZIKV forward: 5′- GGTCAGCGTCCTCTCTAATAAACG-3′; and ZIKV reverse: 5′-GCACCCTAGTGTCCACTTTTTCC-3′, GAPDH forward: 5′-ATGACATCAAGAAGGTGGTG-3′; GAPDH reverse: 5′-CATACCAGGAAATGAGCTTG-3′. USP38 forward: 5′-TCATCAGGAGCCTAACCACC-3′; USP38 reverse: 5′-TCAGGAGAGCAATTACCCACG-3′. 

### 2.6. Lentivirus Production and Infection

The sequence of shRNA targets human USP38 gene (sh-USP38-1) used in this study was as follow: 5′-CGTCTAATACTATGACTGTTA-3′; (sh-USP38-2): CCAGAGATTCTTACTGGTGAT. The shRNA was cloned into pLKO.1 vector (Sigma-Aldrich) and then transfected with psPAX2 and pMD2.G into HEK293T cells. After 36–48 h of transfection, the cell supernatants were collected and filtered through a 0.45-µm filter. Hela cells were infected by collected supernatants for 24 h and 4 μg/mL polybrene (Sigma) was used to assist this infection, after that infected Hela cells were selected by puromycin (Sigma) for 5 days and detected by immunoblot analysis.

### 2.7. Viruses

ZIKV strain z16006 (GenBank accession number, KU955589.1) isolated by the Institute of Pathogenic Microbiology, Center for Disease Control and Prevention of Guangdong (Guangzhou, Guangdong, China) was used in this study. 

### 2.8. Plasmid Construction 

Myc-UB, Myc-K48, Myc-K48R, Myc-K63, and Myc-K63R plasmids were kindly gifted by Dr. Ying Zhu of Wuhan University. The cDNA of human USP38 was provided from Jiahuai Han laboratory of Xiamen University and then cloned into pcDNA3.1 (+)-3× Flag and pCAGGS-HA vectors. The N-terminal domain (1–400aa), C-terminal domain (401–1004aa) of USP38, USP38 (C454A/H857A/D918N) mutant and ZIKV Envelope protein of the corresponding fragment of ZIKV cDNA were cloned into pcDNA3.1 (+)-3 × Flag vector and pCAGGS-HA vector. The expression plasmids Envelope (1–193aa), Envelope (1–296aa), Envelope (1–406aa), Envelope (52–505aa), Envelope (132–505aa), and Envelope (280–505aa) were constructed into pcDNA3.1 (+)-3× Flag vectors.

### 2.9. Statistical Analyses

All experiments are repeated three times independently and the statistical analysis of these experimental data was completed by GraphPad Prism 8 software. Student’s *t*-test was used for analysis between two groups of samples, and one-way ANOVA was used for analysis between multiple groups of samples. *p* < 0.05 was considered statistically significant (ns, there was no significant difference, *: *p* < 0.05, **: *p* < 0.01, and ***: *p* < 0.001).

## 3. Results

### 3.1. USP38 Inhibits ZIKV Infection

Previous studies determined that, during viral infection, USP38 inhibits type I interferon pathway by degrading TBK1, thereby inhibiting the synthesis of type I IFN [15]. Here, we initially explored the role of USP38 in ZIKV infection. The results indicated that, in Hela cells transfected with HA-USP38 and infected with ZIKV, the production of ZIKV structural protein E and non-structural protein 5 (NS5) (Figure 1A), the expression of E mRNA and NS5 mRNA (Figure 1B), and the level of ZIKV mRNA (Figure 1C) were significantly attenuated by USP38. Notably, confocal microscopy showed that the levels of fluorescent stained ZIKV dsRNA were reduced in the presence of HA-USP38 (Figure 1D). These results demonstrate that ZIKV infection was repressed by USP38. To determine the effect of endogenous USP38 on ZIKV infection, short hairpin RNAs (shRNAs) specific target USP38 (sh-USP38) was generated and used to knock-down the endogenous USP38 gene in HeLa cells (Figure 1E). Western blot analyses showed that the productions of ZIKV E protein and NS5 protein were enhanced in the presence of sh-USP38 in ZIKV-infected HeLa cells (Figure 1F). Consistent with this result, the abundance of ZIKV RNA and viral titers was also up-regulated by sh-USP38 in infected cells (Figure 1H,I). To exclude the potential impact of USP38 KD on cell viability, we measured cell viability and, through CCK-8, found out that usp38 knockdown has no significant cytotoxicity (Figure 1J). Similarly, we determined whether USP38 affects ZIKV attachment or invasion. ZIKV was incubated with usp38 knockdown or wild-type Hela cells. q-PCR results showed that knockdown of USP38 did not change ZIKV attachment to cells (Figure 1K). In conclusion, these data proved that USP38 was responsible for the restriction of ZIKV infection.

### 3.2. USP38 Binds to E Protein through Its C-Terminal Domain

Next, the mechanism by which USP38 represses ZIKV infection was explored. HEK293T cells were co-transfected with HA-USP38 and Flag-C, Flag-prM, and Flag-E, respectively. The results of co-immunoprecipitation (Co-IP) showed that USP38 specifically interacted with ZIKV E protein, but failed to interact with ZIKV C protein or prM protein (Figure 2A). Reciprocal Co-IP results further showed that USP38 protein and E protein interacted with each other in HEK293T cells (Figure 2B). The endogenous Co-IP assay showed that, under physiological conditions, the ZIKV E protein can bind to the USP38 protein in ZIKV-infected Hela cells (Figure 2C). In addition, immunofluorescence analyses indicated that USP38 protein and ZIKV E protein co-localized in the cytoplasm of Hela cells (Figure 2D). 

USP38 contains two domains, the N-terminal USP domain and the C-terminal domain containing Ubiquitin carboxyl-terminal hydrolase (UCH). To determine which domain is responsible for the interaction with E protein, we constructed three plasmids, pFlag-USP38 expressing the full-length of USP38, pFlag-USP38-N expressing the N-terminal domain of USP38, and pFlag-USP38-C expressing the C-terminal domain of USP38. Co-IP analysis results showed that E protein interacted with the full-length of USP38 and the C-terminal domain of USP38, but could not interacted with the N-terminal domain of USP38 (Figure 2E). 

ZIKV E protein contains three domains, a central b-barrel (domain I), an elongated finger-like structure (domain II), and a C-terminal immunoglobulin-like module (domain III) [20]. Based on the information of E protein domains, we constructed six corresponding truncated E proteins to investigate which domain of the E protein is critical for the binding of USP38 (Figure 2F). Co-IP results showed that like the full-length E protein, the five truncated E proteins, E (1–193aa), E (1–296aa), E (1–406aa), E (52–505aa), and E (132–505aa), also interacted with USP38, but the truncated protein E (280–505aa) was unable to bind USP38 (Figure 2G,H), indicating that domain I and domain II of the E protein were involved in its binding to USP38. More detailed truncation results showed that USP38 binds to 132–193aa of E protein (Figure 2I). Collectively, our results suggested that USP38 binds to the 132–193aa of E protein through its C-terminal domain.

### 3.3. USP38 Removes Both K48-Linked and K63-Linked Polyubiquitination of E Protein

Previous studies have reported that ZIKV E protein promotes the attachment and entry of the virus into host cells [20], and the activity of E protein is regulated by K63-linked ubiquitination [25]. Here, the role of USP38 in the regulation of E protein ubiquitination was determined. HEK293T cells were co-transfected with Flag-E, Myc-UB, and HA-USP38. The results indicated that USP38 could strongly reduce the ubiquitination level of over-expressed E protein (Figure 3A). In addition, A549 cells were transfected with HA-UB, Flag-vector, or Flag-USP38 for 24 h and then infected with ZIKV (1 MOI). Similarly, the results showed that USP38 also significantly attenuated the ubiquitination level of ZIKV E protein in infected cells (Figure 3B). 

The type of E ubiquitination removed by USP38 was then explored. Interestingly, we noticed that USP38 could remove both K48-linked polyubiquitin and K63-linked polyubiquitin of E protein (Figure 3C). These results were further verified in HEK293T cells transfected with Flag-E, HA-USP38, Myc-UB, Myc-K48R, or Myc-K63R. Notably, ZIKV E protein polyubiquitination was catalyzed significantly by Myc-UB, catalyzed slightly by K48R or K63R, and the levels of E protein polyubiquitination were repressed by USP38 (Figure 3D). The wild-type or USP38 knockdown A549 cells was transfected with HA-UB, and then infected with ZIKV for 48 h. The western blot results showed that the ubiquitin level of E protein increased in the USP38 KD cells (Figure 3E). In summary, USP38 impairs both K48-linked polyubiquitination and K63-linked polyubiquitination of ZIKV E protein.

### 3.4. Deubiquitinase Activity of USP38 Is Required for Restriction of ZIKV Infection

We speculated that deubiquitinase activity of USP38 may play an important role in the restriction of ZIKV infection. To verify this speculation, we constructed a mutant of USP38-MUT (C454A/H857A/D918N) in which the deubiquitinase enzyme activity of USP38 was eliminated. HEK293T cells co-transfected with Flag-E, Myc-UB and HA-USP38 or HA-USP38-MUT. The results showed that ZIKV E protein polyubiquitination catalyzed by Myc-UB were significantly removed by USP38, but relatively unaffected by USP38-MUT (Figure 4A). In addition, Hela cells were transfected with HA-USP38 or HA-USP38-MUT and infected with ZIKV. Immunoblot analyses indicated that the levels of ZIKV E protein and NS5 protein were significantly attenuated by USP38, but not by USP38-MUT (Figure 4B), and similarly, qPCR results revealed that the levels of ZIKV N mRNA and NS5 mRNA (Figure 4C) as well as ZIKV RNA (Figure 4D) were significantly reduced by USP38, but relatively unaffected by USP38-MUT. Furthermore, immunofluorescence analyses indicated that the levels of ZIKV dsRNA were repressed by USP38, but not by the mutant protein (Figure 4E), suggesting that unlike USP38, the mutant protein USP38-MUT failed to repress ZIKV infection. Taken together, these results demonstrate that deubiquitinating activity of USP38 is required for the function of USP38 in the repression of ZIKV infection.

## 4. Discussion

Since the outbreak of Zika virus in South America in 2015, it has aroused public attention [13]. Numerous studies have shown that ZIKV viral infection was responsible for severe neurological effects, such as Guillain-Barré syndrome and microcephaly of newborns [14], and it has also caused testis damage and infertility in male mice [10]. So far, there is still no effective treatment or vaccine for ZIKV infection or ZIKV associated diseases, and mechanistic details regarding innate immunity against ZIKV are still not clear. In this article, we discovered a novel host protein, USP38, which bound to ZIKV E protein and removed polyubiquitination of E protein to resist ZIKV infection.

ZIKV belongs to the genus flavivirus within the family Flaviviridae, and the flavivirus envelope protein is responsible for virus entry and represents a major target of neutralizing antibodies [20]. The post-transcriptional modification of the envelope protein of flavivirus has an important impact on virus transmission, attachment and replication. The glycosylation of envelope protein affects the replication and invasion of viruses such as DENV and ZIKV [26,28,29]. Recent research reported that ZIKV E protein is ubiquitinated by the E3 ubiquitin ligase TRIM7 and this modification improved the efficiency of ZIKV attachment and entry into host cells [25]. Ubiquitin specific peptidase 38 (USP38) was identified as a histone deubiquitinase, and it is a key protein that maintains cell chromatin structure and regulates gene transcription [18,30]. Subsequently, it was revealed that USP38 played a role in virus immunity by inhibiting the IFN-I pathway in the case of virus infection [15]. Here, we demonstrate that, overexpression of USP38 in Hela cells markedly inhibited ZIKV infection, while deficient of USP38 increased ZIKV invasion. We first determined the ability of USP38 to co-immunoprecipitate with three structural proteins (C, prM, and E) of ZIKV, in which E protein scored positive. In the subsequent CO-IP experiment, ZIKV bound to the C-terminal domain of USP38. Additionally, ZIKV E (1–280aa) contained domain I and domain II contributed to the interaction of USP38 and E. Previous studies have shown that USP38 was a deubiquitinating enzyme and we wondered whether USP38 affected the ubiquitination of E protein. We overexpressed USP38 in HEK293T cells and found that USP38 notably reduced the ubiquitination of E protein. Furthermore, USP38 removed both K48- and K63-linked polyubiquitin chains from E. Interestingly, there was no report about the regulation of K48-linked ubiquitination of E, and how this modification affected the replication cycle of ZIKV remained unknown. Ubiquitination modification plays an important role in the replication cycle of flavivirus [31], and the main role of K48-linked ubiquitination is to regulate the stability of the protein by negative feedback [32]. While the virus infects the host, the host cell also has a variety of physiological mechanisms against virus invasion. By adding K48-linked polyubiquitination to viral proteins, the host cell can degrade viral proteins leading to protection against virus infection [33,34]. The K48-linked ubiquitination modification of the ZIKV E protein might be the result of the host’s fight against the virus. USP38 acted as a deubiquitinating enzyme to remove the K48- and K63-linked ubiquitination of E protein. To further prove whether the enzyme activity of USP38 was essential for its inhibition of ZIKV infection; we constructed a USP38 mutant (C454S/H857A/D918N) that has lost deubiquitination enzyme activity. The results revealed that the USP38 (C454S/H857A/D918N) mutant failed to remove polyubiquitination of E, and it also abolished the inhibitory effect of USP38-mediated ZIKV infection. These data suggest that USP38-mediated suppression of ZIKV infection requires its deubiquitinating enzymatic activity.

Based on our data, we proposed the molecular mechanism of USP38 against ZIKV infection. USP38 combines with the central b-barrel (domain I) and elongated finger-like structure (domain II) of E through its C-terminal domain. USP38 was capable to attenuate both K48- and K63-linked ubiquitination of E protein to prevent ZIKV infection. Overall, our study provided new insight into the regulatory mechanisms of Anti-Zika responses and offers a potential target for ZIKV infection.

## Figures and Tables

**Figure 1 viruses-13-02029-f001:**
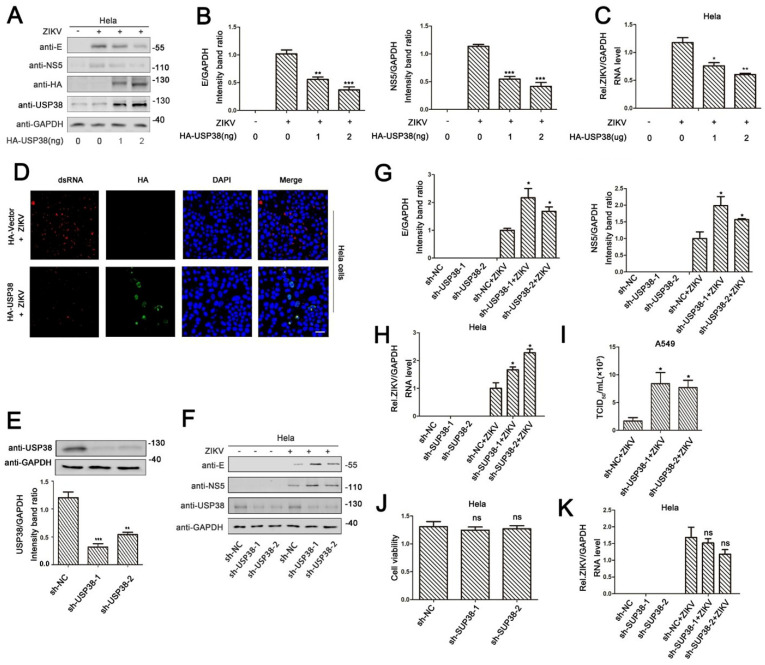
USP38 inhibits ZIKV infection. (**A**–**D**) Hela cells were transfected with HA-USP38 for 24 h then infected with ZIKV (MOI = 1) for 48 h. The levels of ZIKV proteins were detected by immunoblotting (**A**). E/GAPDH and NS5/GAPDH ratio was measured (**B**). The viral RNA content was quantified by qPCR (**C**) and confocal microscopy (**D**), bar = 10 µm. (**E**) Hela cells stably expressing sh-USP38 or control sh-RNA, and the USP38/GAPDH ratio was measured. (**F**–**H**) Hela cells stably expressing sh-USP38 or control sh-RNA were infected with ZIKV (MOI = 1) for 48 h. The expression level of ZIKV proteins were detected by immunoblotting (**F**). The ratios of E/GAPDH (G, left) and NS5/GAPDH (G, right) were measured. The viral RNA content was quantified by qPCR (H). (**I**) A549 cells stably expressing sh-USP38 or control sh-RNA, and the viral titers was measured by TCID_50_. (**J**) Cells were measured for cell viability by CCK-8 assay, and the unit of the y axis is the readout optical density (OD) value. (**K**) Hela cells stably expressing sh-USP38 or control sh-RNA were incubated with ZIKV for 2 h, then continue to be cultured with serum-free medium for 48 h. The viral RNA content was quantified by qPCR. *: *p* < 0.05, **: *p* < 0.01, and ***: *p* < 0.001.

**Figure 2 viruses-13-02029-f002:**
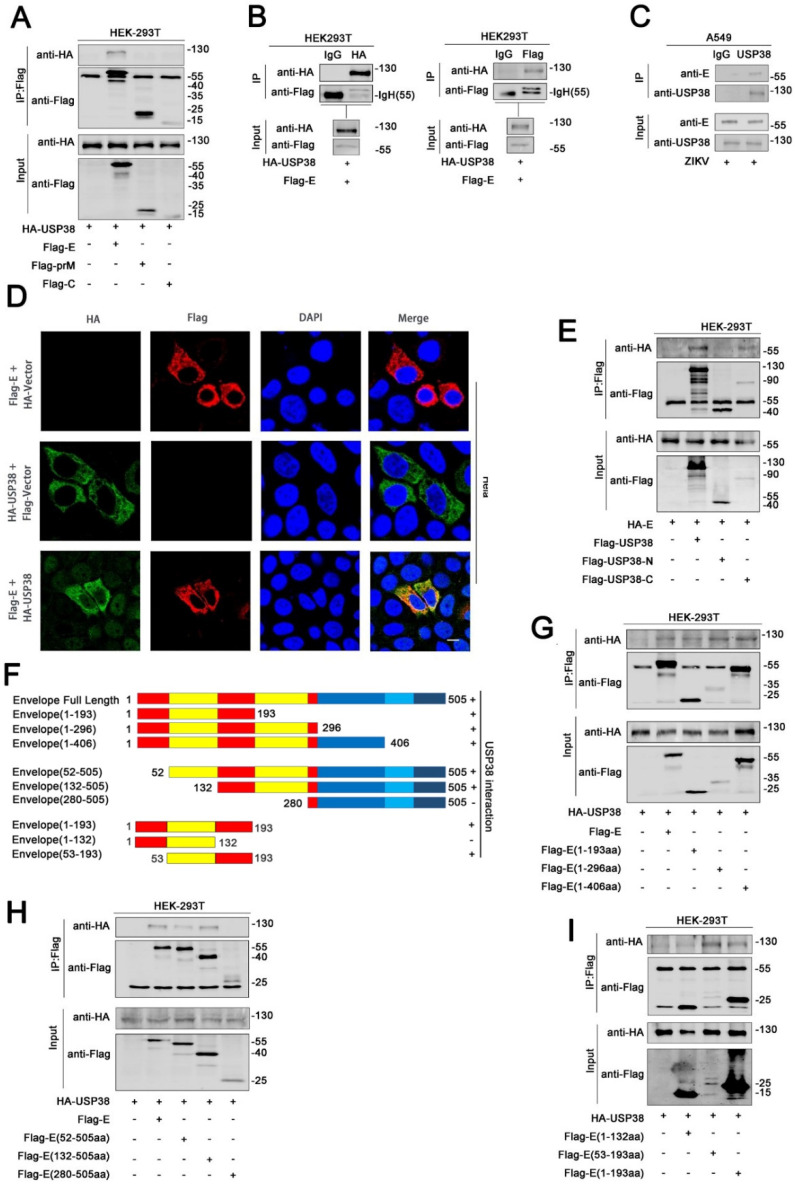
USP38 binds to E protein through its C-terminal domain. (**A**) HEK293T cells were co-transfected with HA-USP38 and Flag-E, Flag-C, Flag-prM, cell lysates were subjected to IP using anti-flag antibody and analyzed by immunoblotting. (**B**) HEK293T cells were co-transfected with HA-USP38 and Flag-E. Cell lysates were subjected to IP using control IgG, anti-HA, or anti-Flag antibody. (**C**) A549 cells was infected with ZIKV for 48 h. Cell lysates were subjected to IP using control IgG or anti-USP38. (**D**) Hela cells were transfected with HA-USP38 or Flag-E, or co-transfected with HA-USP38 and Flag-E. The sub-cellular localizations of HA-USP38 (green), Flag-E (red), and nucleus marker DAPI (blue) were analyzed with confocal microscopy. Bar = 5 µm. (**E**) HEK293T cells were co-transfected with HA-E and Flag-USP38, Flag-N-terminal-USP38 or Flag-C-terminal-USP38. Cell lysates were subjected to IP using anti-Flag antibody and analyzed by immunoblotting. (**F**) Schematic diagram of the full-length E protein and truncated E proteins: E (1–193aa), E (1–296aa), E (1–406aa), E (52–505aa), E (132–505aa) E (280–505aa). (**G**,**H**) HEK293T cells were co-transfected with HA-USP38 and Flag-E, Flag-E (1–193aa), Flag-E (1–296aa), and Flag-E (1–406aa) (**G**) or Flag-E, Flag-E (52–505aa), Flag-E (132–505aa), and Flag-E (280–505aa) (**H**). Cell lysates were subjected to IP using anti-Flag antibody and analyzed by immunoblotting (**G**,**H**). (**I**) HEK293T cells were co-transfected with HA-USP38 and Flag-E (1–193aa), Flag-E (1–132aa), or Flag-E (53–193aa). Cell lysates were subjected to IP using anti-Flag antibody and analyzed by immunoblotting.

**Figure 3 viruses-13-02029-f003:**
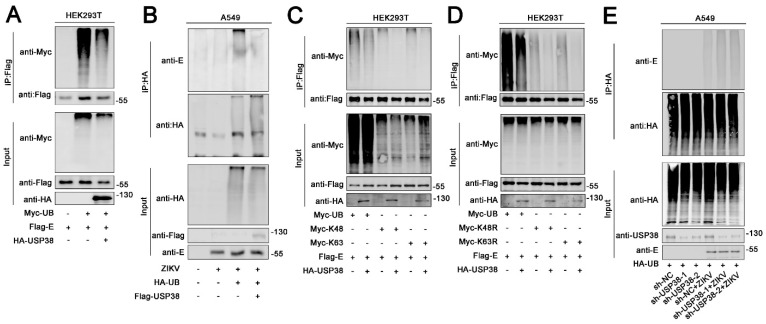
USP38 removes both k48-linked and K63-linked polyubiquitination of E protein. (**A**) HEK293T cells were co-transfected with Flag-E, HA-USP38, and Myc-UB. Cell lysates were immunoprecipitated with anti-Flag and immunoblotted with anti-Myc. (**B**) A549 cells were transfected with HA-UB, Flag-Vector or Flag-USP38 for 24 h then infected with ZIKV (1 MOI). Cell lysates were immunoprecipitated with anti-HA and immunoblotted with anti-E. (**C**) HEK293T cells were co-transfected with Flag-E, Myc-UB, Myc-k48, Myc-k63 together with HA-USP38. Cell lysates were immunoprecipitated with anti-Flag and immunoblotted with anti-Myc. (**D**) HEK293T cells were co-transfected with Flag-E, Myc-UB, MycK48R, Myc-k63R together with HA-USP38. Cell lysates were immunoprecipitated with anti-Flag and immunoblotted with anti-Myc. (**E**) A549 cells stably expressing sh-USP38 or control sh-RNA were transfected with HA-UB for 24h then infected with ZIKV 48h. Cell lysates were immunoprecipitated with anti-HA and immunoblotted with anti-E.

**Figure 4 viruses-13-02029-f004:**
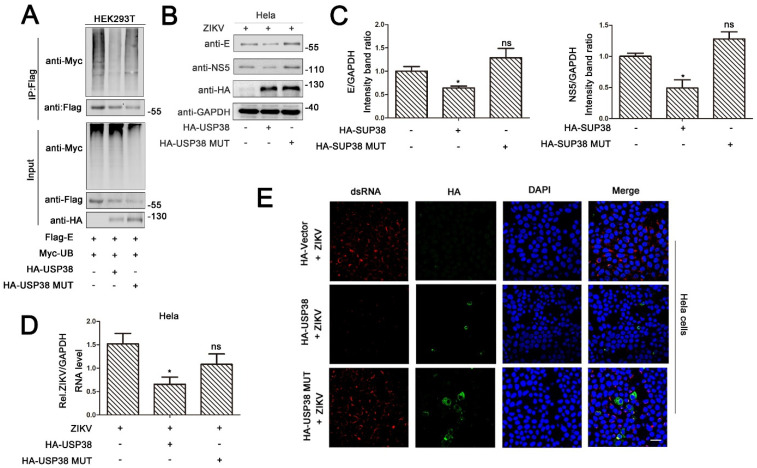
Deubiquitinase activity of USP38 is required for restriction of ZIKV infection. (**A**) HEK293T cells were co-transfected with Flag-E and Myc-UB, together with HA-USP38 or HA-USP38 (C454A/H857A/D918N). Cell lysates were immunoprecipitated with anti-Flag and immunoblotted with anti-Myc. (**B**–**E**) Hela cells were transfected with HA-USP38, HA-Vector or HA-USP38 (C454A/H857A/D918N) for 24 h then infected with ZIKV (MOI = 1) for 48 h. The expression level of ZIKV proteins were detected by immunoblotting (**B**). The ratios of E/GAPDH and NS5/GAPDH ratio were measured (**C**). The viral RNA content was quantified by qPCR (**D**) and confocal microscopy (**E**), bar = 10 µm.

## Data Availability

All the data used in this study is already provided in the manuscript at its required section. There is no underlying data available.

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
