# Peer review of "USP38 Inhibits Zika Virus Infection by Removing Envelope Protein Ubiquitination"

_viruses, 2021, doi:10.3390/v13102029_

Round 1

Reviewer 1 Report

The work by Wang et al., showing that USP38 Inhibits Zika Virus Infection by Removing Envelope protein ubiquitination is overall an interesting study. However,

A general comment is that the authors have not indicated the number of times each experiment was repeated in the manuscript.

  1. Fig 1: The authors must quantify the blot in 1A.
  2. The knockdown efficiency of the  USP38 knock down is not shown. Also, the authors should comment on the viability of the cells after KD of USP38
  3. Fig 2A-2D was performed by over expressing E protein of ZIKV. The authors should performing IP/IF in ZIKV infected cells to support their conclusion that USP38 interacts with E protein of ZIKV.
  4. Fig 2H, the minimum construct that the authors have used is 1-193 aa. Did the authors try to narrow down further this region in E protein that is required for its interaction with USP38. 
  5. Fig 2H, the expression level of Flag-E (280-505aa) in the input is lower as compared to all other truncations. Can the authors rule out that this was not the reason that they could not detect interaction with USP38.
  6. What is the difference between Fig 3C and 3D?
  7. Giraldo et al., reported that ZIKV E protein is ubiquitinated at K63 and this is important for virus entry. Can the authors comment on the role of K48 ubiquitination of E protein.
  8. Since E protein plays a role in virus entry, does knockdown of USP38 makes the cells permissive to the virus. Is there difference in the attachment and entry of viral particles in the cells? 

Minor comments:

  1. Scale bars are missing for microscopy images
  2. Molecular weights should be indicated on the western blot

Reviewer 1 Report

The work by Wang et al., showing that USP38 Inhibits Zika Virus Infection by Removing Envelope protein ubiquitination is overall an interesting study. However, A general comment is that the authors have not indicated the number of times each experiment was repeated in the manuscript.

Authors’ Responses: Thank you for the comments. We have restated in the revised Statistical Analyses that “All experiments are repeated three times independently and the statistical analysis of these experimental data was completed by GraphPad Prism 8 software” in the revised manuscript.

Reviewer 1’ Comments: Fig 1: The authors must quantify the blot in 1A.

Authors’ Responses: Thank you for the comment.

As you suggested, we have quantified the blot of Figure 1A through grayscale analysis by ImagJ, and the new results are shown in the revised Figure 1B of the revised manuscript.

Reviewer 1’ Comments: The knockdown efficiency of the USP38 knock down is not shown. Also, the authors should comment on the viability of the cells after KD of USP38

Authors’ Responses: Thank you for the comment.

As you suggested, we have determined the efficiency of usp38 through grayscale analysis by ImagJ, and the new results are shown in the revised Figure 1E of the revised manuscript.

We have also tested the cell viability of the KD USP38 cell line and WT cell line through the CCK-8 kit, and found that usp38 knockdown has no effect on cell viability. The relevant results are shown in the revised Figure 1J of the revised manuscript.

Reviewer 1’ Comments: Fig 2A-2D was performed by over expressing E protein of ZIKV. The authors should performing IP/IF in ZIKV infected cells to support their conclusion that USP38 interacts with E protein of ZIKV.

Authors’ Responses: Thank you for the comment.

As you suggested, we have performed related experiments to address your question. After 48 hours after Hela cells infected with ZIKV, the cell lysate was IP with control IgG or anti-USP38. The experimental results showed that under physiological conditions, ZIKV E protein can bind to endogenous USP38 protein. The relevant results are shown in the revised Figure 2C of the revised manuscript.

Reviewer 1’ Comments: Fig 2H, the minimum construct that the authors have used is 1-193 aa. Did the authors try to narrow down further this region in E protein that is required for its interaction with USP38.

Authors’ Responses: Thank you for the comment.

As you suggested, we have constructed several new truncated plasmids and found that 132-193aa of ZIKV E protein is the binding domain of USP38 and E. The new results have been shown in the Revised Figure 2I in the revised manuscript.

Reviewer 1’ Comments: Fig 2H, the expression level of Flag-E (280-505aa) in the input is lower as compared to all other truncations. Can the authors rule out that this was not the reason that they could not detect interaction with USP38.

Authors’ Responses: Thank you for the comments.

Regarding this issue, we believed that the Co-Immunoprecipitation experiments are qualitative experiments rather than quantitative experiments, the amount of protein expression would not affect the final result too much. As shown in the revised Figure 2E, the expression levels of the Flag-USP38-C protein and the HA-E protein in the same lane were lower than others, and are more serious than the revised Figure 2H situation, the same situation also occurred in line 4 of the revised Figure 2G, but these did not affect the results of their Co-Immunoprecipitation experiments.

Reviewer 1’ Comments: What is the difference between Fig 3C and 3D

Authors’ Responses: Thank you for the comments.

In Figure 3C, we transfected HA-UB, HA-UB(K48O) (ubiquitin mutant that only retains a single lysine residue) or HA-UB(K63O) (ubiquitin mutant that only retains a single lysine residue). In Figure 3D, we transfected HA-UB, HA-UB(K48R) (only one lysine residue is mutated), or HA-UB(K63R) (only one lysine residue is mutated). The results showed that USP38 decreased K48- and K63-linked polyubiquitination of ZIKV E protein. We combined the results of these two pictures and got the conclusion that USP38 decreased K48- and K63-linked polyubiquitination of ZIKV E protein.

Reviewer 1’ Comments: Giraldo et al., reported that ZIKV E protein is ubiquitinated at K63 and this is important for virus entry. Can the authors comment on the role of K48 ubiquitination of E protein.

Authors’ Responses: Thank you for the comment.

As you suggested, the role of K48 ubiquitination of E protein have been illustrated in the discussion section of the revised manuscript.

Reviewer 1’ Comments: Since E protein plays a role in virus entry, does knockdown of USP38 makes the cells permissive to the virus. Is there difference in the attachment and entry of viral particles in the cells?

Authors’ Responses: Thank you for the comment.

As you suggested, we have performed related experiments to address your question. The new results have been shown in the revised Figure 1K in the revised manuscript.

Reviewer 1’ Minor comments:

  1. Scale bars are missing for microscopy images
  2. Molecular weights should be indicated on the western blot

Authors’ Responses: Thank you for the comment.

We have added Scale bars are missing for microscopy images and Molecular weights should be indicated on the western blot in the revised manuscript.

Reviewer 2 Report

In this manuscript, Wang et al. asses the role of the deubiquitinase USP38 in Zika virus infection. Whilst this could be potential interesting as deubiquitinases are an attractive therapeutic target in a range of diseases (including viral infections), this work is currently not of sufficient strength to support the conclusions presented.

  • Figure 1A - please provide a USP38 blot.
  • Figure 1A - Quantification for NS5 and E protein expression from at least three repeats should be provided. 
  • Figure 1 - Zika virus titres should be assessed, not just protein expression and RNA levels, as this will give a more representative assessment of the role of USP38 on infection.
  • Figure 1B - is this negative or positive strand? Both should be assessed.
  • Figure 1C - how is USP38 reducing dsRNA levels in cells not expressing HA-USP38? This must be addressed.
  • Figure 1D and E - a single shRNA is not sufficient; at least two shRNAs targetting different regions must be used.

  • Figure 2 - does E protein bind to endogenous USP38? This should be assessed.

  • Figure 3 - USP38 depletion using 2 shRNAs should be used to assess if E protein ubiquitination increases during infection.

  • Figure 4 - as for Figure 1, viral titre and both negative and positive strand RNA levels should be assessed.
  • Figure 4D - as with Figure 1, how is USP38 reducing dsRNA levels in cells not expressing HA-USP38? This must be addressed.

Reviewer 2 Report

In this manuscript, Wang et al. asses the role of the deubiquitinase USP38 in Zika virus infection. Whilst this could be potential interesting as deubiquitinases are an attractive therapeutic target in a range of diseases (including viral infections), this work is currently not of sufficient strength to support the conclusions presented.

Author Response: Thank you for the commends.

    We have improved and strengthen conclusions by providing more evidence in the revised manuscript.

Reviewer 2’ Comments: Figure 1A - please provide a USP38 blot.

Authors’ Responses: Thank you very much for your suggestion.

The ‘anti-HA’ blot in Figure 1A is the expression band of HA-USP38.

Reviewer 2’ Comments: Figure 1A - Quantification for NS5 and E protein expression from at least three repeats should be provided.

Authors’ Responses: Thank you very much for your suggestion.

As you suggested, we have counted the results of three experiments and performed grayscale analysis. The new results have been shown in the revised Figure 1B in the revised manuscript.

Reviewer 2’ Comments: Figure 1 - Zika virus titer should be assessed, not just protein expression and RNA levels, as this will give a more representative assessment of the role of USP38 on infection.

Authors’ Responses: Thank you very much for your suggestion.

As you suggested, we have conducted TCID50 experiments to measure the titer of ZIKV to evaluate the impact of USP38 on ZIKV infection. The new results have been shown in the revised Figure 1I in the revised manuscript.

Reviewer 2’ Comments: Figure 1B - is this negative or positive strand? Both should be assessed.

Authors’ Responses: The ZIKV primers used in this article is a positive-strand primer (software: DNAMAN), as shown in the figures below.

The replication cycle of Zika virus is the same as other flaviviruses.

‘A (+) strand genomic RNA serves as a template to produce (−) strand RNA. The (−) strand RNA exists as a dsRNA intermediate (replicative form). The (−) strand within the dsRNA intermediate is then used as a template for (+) strand RNA synthesis. The dsRNA product is released and recycled for additional (+) strand synthesis.’ [Klema, V. J., R. Padmanabhan and K. H. Choi (2015). "Flaviviral replication complex: coordination between RNA synthesis and 5’-RNA capping." Viruses 7(8): 4640-4656.]

From the replication cycle of Zika virus, we can know that measuring the RNA content of the positive strand is sufficient to verify the replication of Zika virus. The same results can be obtained through the TCID50 experiments and the Western Blot experiments.

Reviewer 2’ Comments: Figure 1C - how is USP38 reducing dsRNA levels in cells not expressing HA-USP38? This must be addressed.

Authors’ Responses: Thank you very much for your suggestion.

In this experiment, the HA-USP38 were overexpressed in Hela cells, and these proteins could reduce the ubiquitination level of the E protein translated in the host cells. The assembled Zika virus, due to the low level of E protein ubiquitination, the ability to infect the host cell decreases. Although some cells did not overexpress the USP38 protein, the overall infectivity of the virus decreased, and the virus is not easy to infect these cells,which made the dsRNA fluorescence decrease in cells without green fluorescence.

Reviewer 2’ Comments: Figure 1D and E - a single shRNA is not sufficient; at least two shRNAs targetting different regions must be used.

Authors’ Responses: Thank you very much for your suggestion.

As you suggested, we have designed new shRNA sequence and completed related experiments. The new results are shown in the revised Figure 1F and the revised Figure 1H in the revised manuscript.

Reviewer 2’ Comments: Figure 2 - does E protein bind to endogenous USP38? This should be assessed.

Authors’ Responses: Thank you for the comment.

As you suggested, we have performed related experiments to address your question. After 48 hours after Hela cells infected with ZIKV, the cell lysate was IP with control IgG or anti-USP38. The experimental results showed that under physiological conditions, ZIKV E protein can bind to endogenous USP38 protein. The relevant results are shown in the revised Figure 2C of the revised manuscript.

Reviewer 2’ Comments: Figure 3 - USP38 depletion using 2 shRNAs should be used to assess if E protein ubiquitination increases during infection.

Authors’ Responses: Thank you very much for your suggestion.

As you suggested, we have performed related experiments to address your question. The new results are shown in the revised Figure 3E in the revised manuscript.

Reviewer 2’ Comments: Figure 4 - as for Figure 1, viral titre and both negative and positive strand RNA levels should be assessed.

Authors’ Responses: Thank you very much for your suggestion.

Because the TCID50 experiment requires 5-7days to culture a549 cells, but the expression of overexpressed plasmids will be greatly reduced after a long time (more than 48 hours), resulting in the disappearance of the phenomenon, so we failed to measure the virus titer in Figure 4.

Reviewer 2’ Comments: Figure 4D - as with Figure 1, how is USP38 reducing dsRNA levels in cells not expressing HA-USP38? This must be addressed.

Authors’ Responses: Thank you very much for your suggestion.

In this experiment, the HA-USP38 were overexpressed in Hela cells, and these proteins could reduce the ubiquitination level of the E protein translated in the host cells. The assembled Zika virus, due to the low level of E protein ubiquitination, the ability to infect the host cell decreases. Although some cells did not overexpress the USP38 protein, the overall infectivity of the virus decreased, and the virus is not easy to infect these cells, which made the dsRNA fluorescence decrease in cells without green fluorescence.

Round 2

Reviewer 1 Report

The authors have addressed all my queries.

Minor comments:

  1. In Figure 1B (quantification of blot in 1A), the authors should mention on the which samples are infected and which ones are not infected (like they do in 1C).
  2. The authors should show 132-193aa domain interaction in 2I

Author Response

Reviewer 1 Comment: The authors have addressed all my queries.

Authors’ Responses: Thank you very much for your comment.

Reviewer 1 Minor comments:

  1. In Figure 1B (quantification of blot in 1A), the authors should mention on the which samples are infected and which ones are not infected (like they do in 1C).
  2. The authors should show 132-193aa domain interaction in 2I.

Authors’ Responses: Thank you for your suggestions.

As you suggested, we have resolved these issues in the revised manuscript.

Reviewer 2 Report

The authors have addressed most of my comments.

However, in Figure 1A the HA blot may demonstrate expression of the HA-USP38 construct, but a USP38 blot would provide more information, demonstrating the expression of the protein of interest. This should be easy for the authors to do as they have other USP38 blots in the manuscript.

Author Response

Reviewer 2 Comment: The authors have addressed most of my comments.

Authors’ Responses: Thank you very much for your comment.

Reviewer 2 Comment: However, in Figure 1A the HA blot may demonstrate expression of the HA-USP38 construct, but a USP38 blot would provide more information, demonstrating the expression of the protein of interest. This should be easy for the authors to do as they have other USP38 blots in the manuscript.

Authors’ Responses: Thank you for the comment.

As you suggested, we have repeated the Western blot experiments and provided a USP38 blot in the revised Figure 1A in the revised manuscript.
